# Past, present and future in the geographical distribution of Mexican Tepezmaite cycads: Genus *Ceratozamia*

**Jorge Antonio Gómez-Díaz[1,2], César Isidro Carvajal-Hernández[2]\*, Wesley Dáttilo[3]**

**1** Centro de Investigaciones Tropicales, Universidad Veracruzana, Xalapa, Veracruz, México, **2** Instituto de Investigaciones Biológicas, Universidad Veracruzana, Xalapa, Veracruz, México, **3** Red de Ecoetología, Instituto de Ecología A.C., Xalapa, Veracruz, México

\* ccarvajal@uv.mx

**Data Availability Statement:** All the files used to obtain the results are available in the GBIF public access databases: GBIF.org. The Global Biodiversity Information Facility; 2023 [cited 21

## Abstract

*Ceratozamia morettii*, *C. brevifrons*, and *C. tenuis* are cycads considered endangered in montane forests in the center of Veracruz state. However, the amount of theoretical and empirical information available on the historical distribution of these species and how they could be affected in the future by the effects of climate change still needs to be increased. Our objective was to generate information on the spatial distribution of the species since the last glacial maximum, present, and future. To map the spatial distribution of species, we created a potential distribution model for each species. The spatial data used for the models included 19 bioclimatic data variables in the present, at the last glacial maximum using two models (CCSM4 and MIROC), and in the future (2080) using two models of the RCP 8.5 scenario of climate change (HadGEM2-CC and MIROC5). We found that each species occupies a unique ecoregion and climatic niche. *Ceratozamia morettii* and *C. tenuis* have a similar pattern with an expansion of their distribution area since the last glacial maximum with a larger distribution area in the present and a projected reduction in their distribution under future climatic conditions. For *C. brevifrons*, we also showed an increase in their distributional area since the last glacial maximum. We also showed that this expansion will continue under future climatic conditions when the species reaches its maximum distributional area. Projections about the future of these endemic cycad species show changes in their habitat, highlighting that temperate zone species (*C. morettii* and *C. tenuis*) will face imminent extinction if no effort is made to protect them. On the other hand, the tropical climate species (*C. brevifrons*) will be favored.

## Introduction

Cycads are an endangered plant group distributed in tropical and subtropical regions [1]. They are considered living fossils and evolved from the ancient "seed ferns" of the late Paleozoic. There are 375 described species of cycads in the world [2]. In Mexico, 74 species of cycads are recognized in three genera: *Ceratozamia*, *Dioon*, and *Zamia* [2]. Of these species, 80% are

**Funding:** The research was supported by Mohamed bin Zayed Species Conservation Fund (project number 192521089) and also the Centro de Investigaciones Tropicales, Universidad Veracruzana for their support for conducting fieldwork. The funders had no role in study design, data collection and analysis, decision to publish, or preparation of the manuscript.

endemic to Mexico, making it the second most diverse country in the world regarding cycad diversity, after only Australia [3].

Cycads live in tropical and subtropical environments, from humid jungles, dry jungles, mountain cloud forests, pine-oak forests, and scrublands [3, 4]. Unlike other non-flowering plants, cycads are pollinated by primitive insects (Coleoptera and Thysanoptera). Some have their origin in the mid-Cretaceous [5, 6]. The seeds of cycads have been used to supplement people's diet in times of scarcity, but in Mexico, their main use is as ornamental; the leaves of some cycads are used to decorate churches during religious festivals [7]. Many species are removed from their natural habitats by collectors, and plant traders [3], added to the reduction of their habitats has caused that currently more than 60% of cycad species in the world to be threatened [4].

Among the cycads is the *Ceratozamia* genus (Zamiaceae), which is exclusively distributed in the neotropics [8, 9]. Most of the species of *Ceratozamia* are in the mountainous regions of Mexico, specifically on the mountain range of the Sierra Madre Oriental, the area with the highest diversity of this genus [8, 9]. Within the genus, *Ceratozamia* are *C. brevifrons* (Fig 1A), *C. morettii* (Fig 1B), and *C. tenuis* (Fig 1C), three micro endemic species from central Veracruz, known collectively by the local name "Tepezmaite" or "Palmita," which are at risk of disappearance [7]. All three species are considered endangered (EN) by International Union for Conservation of Nature (IUCN) and are listed in section I of CITES [10, 11]. On the other hand, *C. morettii* and *C. brevifrons* (the latter as synonyms are *C. mexicana*) are also protected by Mexican laws [12, 13]. Mexican laws do not protect *C. tenuis* due to its recent lectotype assignment in 2016 [14], previously considered a synonym of *C. mexicana*. *Ceratozamia morettii* and *C. tenuis* grow in mountain cloud forests, while *C. brevifrons* grow in tropical oak forests; both environments are among Mexico's most threatened types of vegetation [15]. Besides, the habitat has strong pressures caused by agricultural and livestock activities [16]. Therefore, these species are particularly interesting for conservation strategies and management purposes.

In general, IUCN range polygons are frequently used to make decisions for species conservation [17, 18]. However, this methodology does not consider the effects of environmental variables that could influence the species and sometimes can severely underestimate real-world occupancy [19–21]. Therefore, it has been recognized that the IUCN range polygons are imprecise and overestimate the size of the distribution of the species, with the consequences for their conservation that this methodology entails [22]. Species distribution modeling has been used to determine the suitability of the species' ecological niche (i.e., species habitat

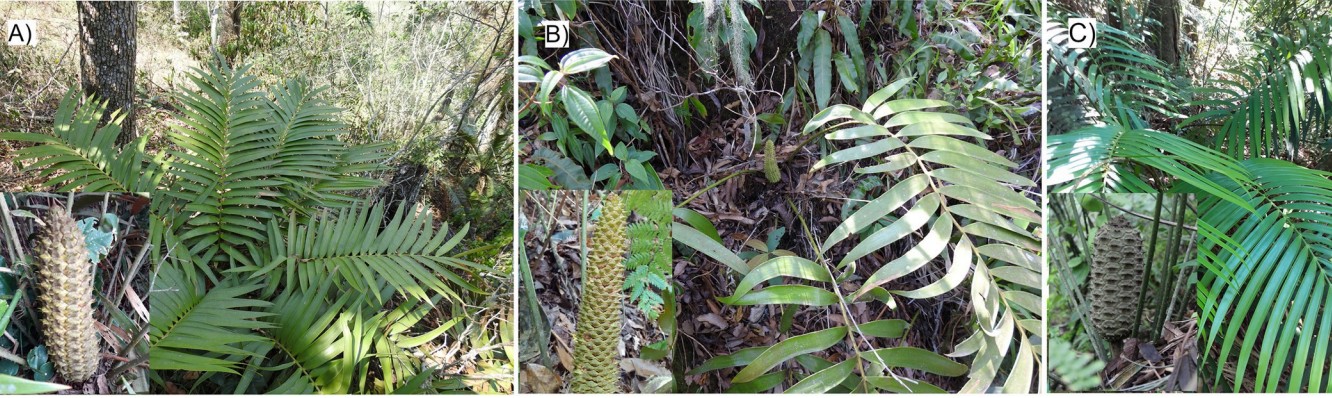

**Fig 1. Tepezmaite cycads.** a) *Ceratozamia brevifrons* with female cone; b) *C. morettii* with male cone; c) *C. tenuis* with female cone.

requirements necessary for long-term population persistence). This is because niche ecological models were developed to predict species distributions from presences [23], which, together with the ease and progress made in this area, has led to their greater use.

In addition to the geographic information provided by species distribution modeling, it has also been used to conserve different taxonomic groups such as (birds, mammals, and plants). However, this tool has yet to be used for cycads since only some works address these species from the perspective of their conservation [24]. The evaluation of the species concerning the past conditions in the glaciations allows us to understand how the contractions in the distribution of the species occurred under adverse weather conditions [25]. In this sense, in old groups such as the cycads, which come from more than 200 million years [26], the information generated by species distribution modeling can help us to understand know how they developed in conditions different from the current ones in addition to the fact that currently, most species are in danger of extinction [4].

In general, the species that inhabit mountainous areas, such as the Tepezmaite's cycads, are the most vulnerable to the effects of climate change [27, 28]. The main impact of climate change on these species is found in the movement of their populations since they seek their preferred climatic niches over time, which is why montane populations migrate to new higher areas to avoid changes in the climate [28, 29]. However, the high rates of deforestation and transformation of montane habitats wreak havoc on the survival of populations at ever-increasing rates; if climate change is added to this, populations are increasingly fragmented and at greater risk of extinction [28, 30].

Therefore, one of the goals of the species distribution models is to prevent and forecast changes in the distribution patterns of threatened species in the face of the threat of climate change, with which the necessary actions could be taken to help in their conservation [28, 31]. Unfortunately, it is not known what the effects of climate change could be on the distribution of Tepezmaite cycads, which is one more threat to the conservation of these species [27]. In fact, according to the IUCN, conducting assessments regarding the effect of climate change on species is important to anticipate the effects that the future climate may have on current populations and thus expect and act effectively for their conservation [32].

Therefore, the objective of this study was to know the changes in the past distribution (last glacial maximum), present and future (under the effects of climate change) of the Tepezmaite cycads (*C. brevifrons*, *C. morettii*, and *C. tenuis*), to understand what is the history that the species have had and how they could be affected in the future by the effects of climate change. Specifically, we seek to know: i) which are the environmental variables that most influence the model of the ecological niche of each species, ii) which was the distribution of each species in the last glacial maximum, iii) which is the present distribution of each species and iv) what are the effects of climate change on species for the year 2080.

## Methods

### Study area

Our study area lies in the center of Veracruz state, Mexico, in the Sierra de Chiconquiaco, where the three micro endemic species are located (Fig 2). The Sierra de Chiconquiaco in the Sierra Madre Oriental has a complex micro-environment system and great cultural wealth (Fig 2). This is due to the short distance between the coastal zone and the elevations of the mountain range, which rises from thirty-five to almost 3,000 m a.s.l. [33]. The biodiversity of this area is threatened due to several activities, such as deforestation, forest fires, livestock, and agriculture, particularly in the mountains [34]. This area is topographically and climatically complex, with seven types of vegetation. It possesses a high plant diversity with 3019 species,

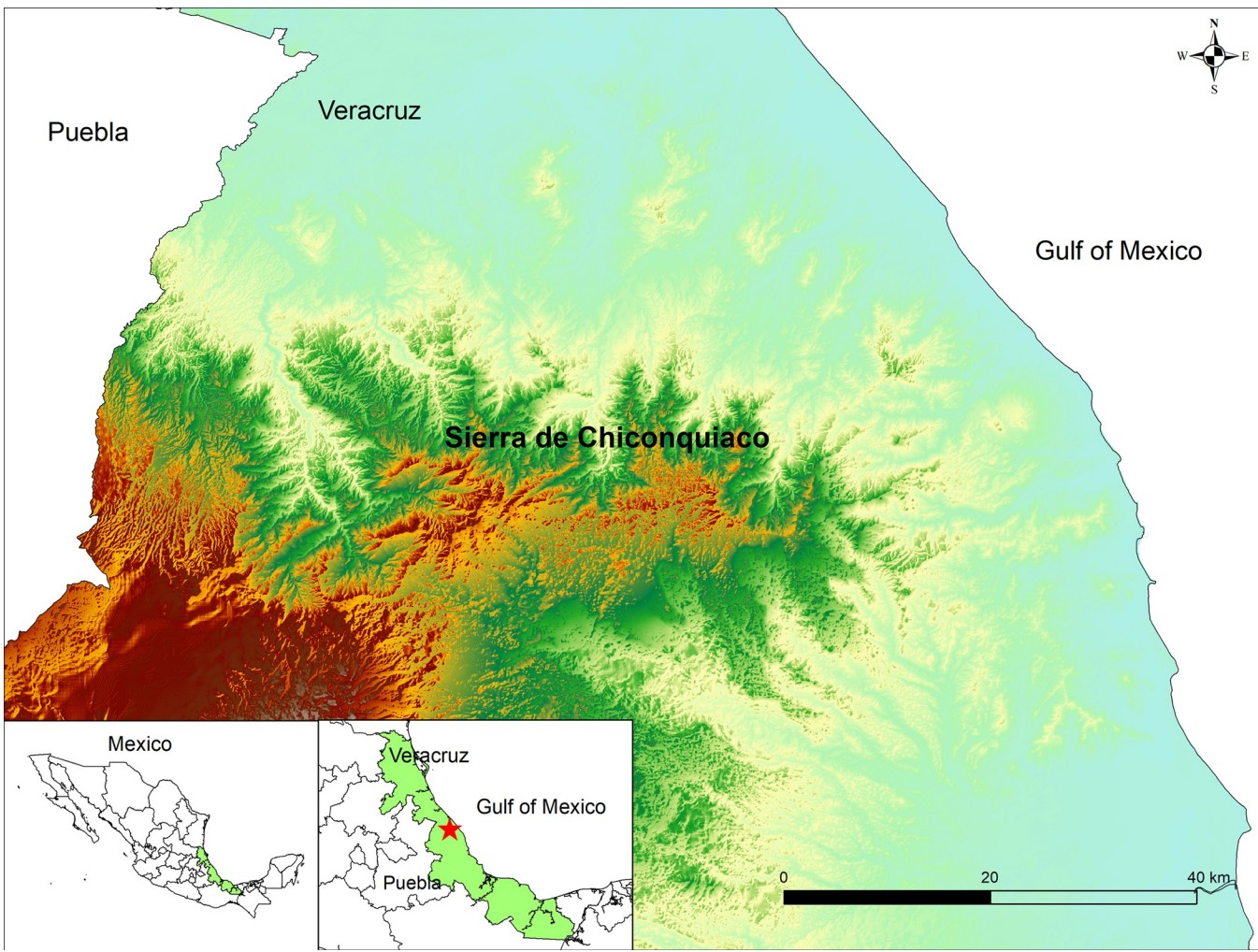

**Fig 2. Study area in the Sierra de Chiconquiaco, Veracruz, Mexico.**

including thirty-six endemic, 57 protected by Mexican laws, and 195 listed in CITES (including six cycads) [33].

## Mapping the species

To map the spatial distribution of species, we obtained records of the studied *Ceratozamia* species from our data in the field, data from the XAL, XALU, and MEXU herbaria, data from the literature and the Global Biodiversity Information Facility (GBIF), as well as corroboration of localities in the field. Only georeferenced records were chosen. We found 14 occurrences for *C. brevifrons* [35], 16 for *C. morettii* [36] and 26 for *C. tenuis* [37]. The coordinates were cleaned up (i.e., doubtful records based on known distributional ranges) using the *clean_coordinates* function from the *CoordinateCleaner* R package [38]. The data was cleaned to have only one presence per km$^2$. In the end, only 13 records of *C. brevifrons* remained, 16 of *C. morettii*, and 21 of *C. tenuis* entered the distribution model. Once the database with the points of presence was completed, a potential distribution model for each species was conducted.

The spatial data used for modeling included nineteen bioclimatic data variables from the CHELSA v.1.2 database at a resolution of 30 arc seconds [39]. The CHELSA bioclimatic

variables are the most suitable in mountainous and tropical areas [40]. For the present, the last glacial maximum (LGM; ~ 22,000 years ago; past henceforth), and the scenario of future climate change (2080; future subsequently) projections. We considered the same bioclimatic variables for the distribution modeling in the past, present, and future scenarios.

In the case of the past, we used the median of the predictions of two Global Circulation Models (CGMs) models (CCSM4 and MIROC). In the case of the future, we also used the median of two CGMs models (HadGEM2-CC and MIROC5), which are commonly used and recommended for Mexico based [41] on the RCP (Representative Concentration Pathway) 8.5 scenario of climate change of the IPCC (Intergovernmental Panel on Climate Change). We only considered the RCP 8.5 (936 ppm of $CO_2$) scenario because it represents a continuous increase in global emissions throughout the 21$^{st}$ century and appears to be the most likely scenario, given the steady trend so far of greenhouse effect gas emissions [42]. Therefore, if any species could survive in this scenario is highly likely to endure future climate outcomes.

An area of exploration of the variables (M) was calibrated using the terrestrial ecoregions of The Nature Conservancy based on those created by [43]. The ecoregions with at least one presence record were selected for each species. The variables were trimmed to the M of each species. Also, the environmental variables were revised to avoid multiple collinearities between the variables using the variance inflation factor (VIF) analysis. Variables with a VIF value > 10 were eliminated using the exclude function from the R usdm version 1.1–18 package [44]. VIF indicates the degree to which standard errors are inflated due to levels of multicollinearity. VIF values > 10 were taken as indicative of problematic collinearity/redundancy [45] since high collinearity could lead to poor model performance and misinterpretations [40]. Of the 19 variables, only six variables did not present multicollinearity (VIF <10) for *C. brevifrons* and five for both *C. morettii* and *C. tenuis* (Table 1).

For each species, an Ensemble species distribution model (ESDM) was generated, assembling all the methods available in the R package *SSDM* using the function *ensemble_modelling*. The methods included: Generalized linear model (GLM), Generalized additive model (GAM), Multivariate adaptive regression splines (MARS), Generalized boosted regressions model (GBM), Classification tree analysis (CTA), Random forest (RF), Maximum entropy (MAXENT), Artificial neural network (ANN), and Support vector machines (SVM). To generate the binary map of each species, the sensitivity-specificity equality (SES) metric was chosen [46]. Subsequently, AUC (area under the curve) values were calculated to describe the model performance or predictive accuracy. This study performed all modeling analyses using the programming language R v.3.6.3 [47].

**Table 1. Bioclimatic variables (names and units) used as predictors in the species distribution models of Tepezmaite cycads (*Ceratozamia brevifrons*, *C. morettii*, and *C. tenuis*).**

| Variable | Name | Species |
|---|---|---|
| Bio 2 | Mean diurnal range | *Ceratozamia brevifrons* |
| Bio 3 | Isothermality | *Ceratozamia morettii* |
| Bio 4 | Temperature seasonality | *Ceratozamia brevifrons*, *C. morettii* |
| Bio 7 | Temperature annual range | *Ceratozamia tenuis* |
| Bio 8 | The mean temperature of the wettest quarter | *Ceratozamia morettii* |
| Bio 9 | The mean temperature of the driest quarter | *Ceratozamia brevifrons* |
| Bio 11 | The mean temperature of the coldest quarter | *Ceratozamia brevifrons*, *C. tenuis* |
| Bio 13 | Precipitation of the wettest month | *Ceratozamia brevifrons*, *C. morettii*, *C. tenuis* |
| Bio 14 | Precipitation of the driest month | *Ceratozamia tenuis* |
| Bio 18 | Precipitation of the warmest quarter | *Ceratozamia brevifrons*, *C. morettii*, *C. tenuis* |

**Table 2. The threshold to categorize the suitability and evaluation metrics for the ESDM for each *Ceratozamia* studied.**

| Metric | C. brevifrons | C. morettii | C. tenuis |
|---|---|---|---|
| Threshold | 0.629 | 0.621 | 0.797 |
| AUC | 0.951 | 0.919 | 0.985 |
| Omission rate | 0.127 | 0.131 | 0.102 |
| Sensitivity | 0.945 | 0.914 | 0.986 |
| Specificity | 0.870 | 0.859 | 0.896 |
| The proportion of correctly predicted occurrences | 0.873 | 0.869 | 0.898 |
| Kappa | 0.582 | 0.521 | 0.744 |
| Calibration | 0.587 | 0.550 | 0.681 |

## Results

In general, all the models for the *Ceratozamia* species have an AUC above 0.9, which means that each model's prediction was accurate (Table 2). Despite the three species living extremely near each other (~10 km) and having an exceptionally small distribution area in central Veracruz (Fig 3), each occupies a unique ecoregion and climatic niche. In this case, the ESDM for each species shows that they have a difference in the importance of bioclimatic variables influencing the distribution of the *Ceratozamia*. In the case of *C. brevifrons*, the most important variables were temperature seasonality (Bio 4,41%), mean diurnal range (Bio2,17%), and precipitation of wettest month (Bio 13,16%); for *C. morettii*, the most important variables were mean temperature of wettest quarter (Bio 8, 29%), Bio 13 (24%) and Bio 4 (23%), and for *C. tenuis* were precipitation of driest month (Bio 14,29%), mean temperature of coldest quarter (Bio 11,24%) and precipitation of warmest quarter (Bio 18,20%; Table 3). The only variables shared for the three species are Bio 13 and Bio 18.

### Ceratozamia brevifrons

The past model suggested a slight decrease regarding the present distribution (Table 4) of occurrence for *C. brevifrons*, and the species was restricted to the area's mountains (Fig 4A). The species increased by 155% in geographical distribution from the past to the present. The present potential distribution of *C. brevifrons* covered the area of central Veracruz, with the most favorable conditions on the east of the Sierra de Chiconquiaco in some mountains and near the coast (Table 4, Fig 4B). This species has the lowest elevation range of the three species studied (500 m a.s.l.). Less favorable areas for *C. brevifrons* were defined in the forests at the north and south of the Sierra de Chiconquiaco. In the future scenario, an increase of 60% in the species' geographical range is expected (Tables 4 and 5, Fig 4C).

### Ceratozamia morettii

The past model suggested a decrease in occurrence for *C. morettii* (Table 4, Fig 4D). The species increased by 985% in its geographical range from the past to the present (Table 5). The present potential distribution of *C. morettii* covered the area of central Veracruz, with the most favorable conditions in the Sierra de Chiconquiaco (Table 4, Fig 4E). In this area, there are a large number of records of the species. The least favorable areas for *C. morettii* were found in the mountain cloud forest at the south and west of the Sierra de Chiconquiaco. But, under the climate change scenario of the models HadGEM2-CC and MIROC 5, a reduction of 88% in the species' distribution range is expected (Tables 4 and 5, Fig 4F). Except for limited high elevated areas in the mountains of the Sierra de Chiconquiaco, the species practically disappears.

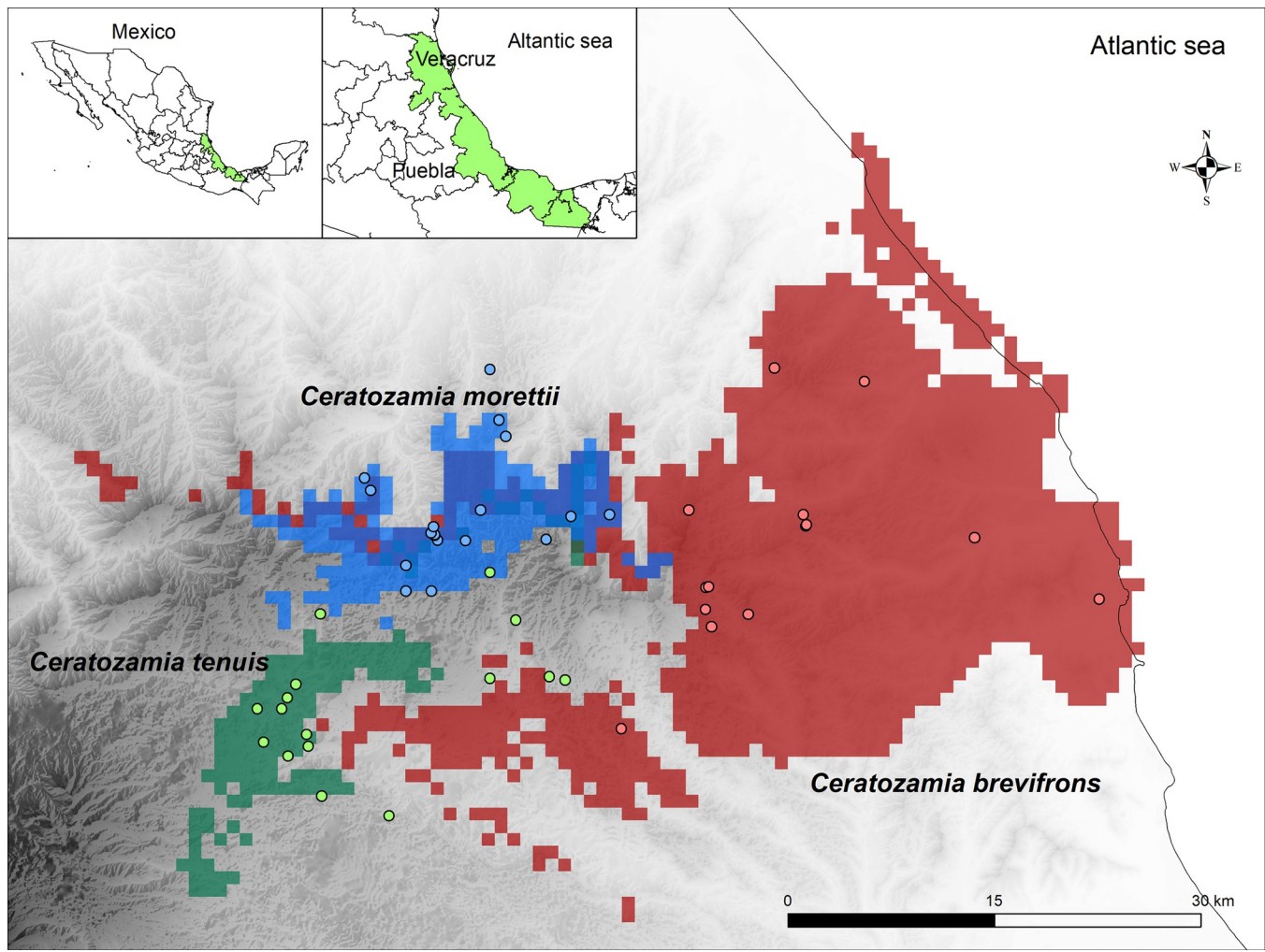

**Fig 3. Potential distribution of the studied *Ceratozamia* species with the locations of occurrence of each species (points) in the Sierra de Chiconquiaco, Veracruz, Mexico; the grey shadow represents topographic relief.**

**Table 3. Variable importance (%) for the ensemble species distribution model of each *Ceratozamia* studied.**

| Variable | Name | *C. brevifrons* | *C. morettii* | *C. tenuis* |
|---|---|---|---|---|
| Bio 2 | Mean diurnal range | 17 | | |
| Bio 3 | Isothermality | | 14 | |
| Bio 4 | Temperature seasonality | 41 | 23 | |
| Bio 7 | Temperature annual range | | | 13 |
| Bio 8 | The mean temperature of the wettest quarter | | 29 | |
| Bio 9 | The mean temperature of the driest quarter | 7 | | |
| Bio 11 | The mean temperature of the coldest quarter | 8 | | 24 |
| Bio 13 | Precipitation of the wettest month | 16 | 24 | 15 |
| Bio 14 | Precipitation of the driest month | | | 29 |
| Bio 18 | Precipitation of the warmest quarter | 11 | 10 | 20 |

**Table 4. Projected suitability area (km$^2$) of the three species of *Ceratozamia* studied in the various times past (last maximum glacial), present, and future (2080).**

| Time | Suitability | *C. brevifrons* | *C. morettii* | *C. tenuis* |
|---|---|---|---|---|
| Past | High | 5 | 0 | 0 |
| | Medium | 1409 | 41 | 74 |
| Present | High | 1096 | 166 | 121 |
| | Medium | 2515 | 279 | 2917 |
| Future | High | 2214 | 0 | 0 |
| | Medium | 3581 | 54 | 529 |

### Ceratozamia tenuis

The past model suggested a decrease in occurrence for *C. tenuis* (Table 4, Fig 4G). The species increased by 4005% in its geographical range from past to present (Table 5). The present potential distribution of *C. tenuis* covered the area of central Veracruz, with the most favorable conditions in the mountain cloud forests in the south of the Sierra de Chiconquiaco (Table 4, Fig 4H). In this area, there are a large number of records of the species. The least favorable areas for *C. tenuis* were found in the mountain cloud forest south and west of the Sierra de Chiconquiaco. But, under the climate change scenario, a reduction of 83% of the species distribution range is expected (Tables 4 and 5, Fig 4I). Except for limited areas in mountainous areas on the south of the Sierra de Chiconquiaco, the species practically disappeared.

## Discussion

We estimated the past, present, and future distribution of three endemic cycads of Veracruz (*C. brevifrons*, *C. morettii*, and *C. tenuis*). We provided a quantitative assessment of the Spatio-temporal changes in the potential distribution of these threatened species since the LGM and as a consequence of global climate change. In summary, we found: (a) a general increase in the distributional area of the species since the LGM, (b) a decrease in the range occupied by *C. morettii* and *C. tenuis* under future scenarios, and (c) an increase in the range occupied by *C. brevifrons* in future scenarios.

We found that the *Ceratozamia* had dynamics in their distribution range with expansions from the LGM. The modeled distributions showed a spatial separation among the three species since the LGM, a pattern also found for Mexico's conifer *Podocarpus* (Podocarpaceae) [48]. This could be explained by the difference between the bioclimatic variables that affect each species. The models indicate that the precipitations of the wettest month (Bio 13) and the warmest quarter (Bio 18) were the only variables shared for the three species of *Ceratozamia*. Moreover, we found that the temperature seasonality (Bio 4) was important for *C. brevifrons*, while the average temperature of the wettest quarter (Bio 8) was for *C. morettii* and precipitation of the driest month (Bio 14) was for *C. tenuis*. These findings are accordingly to [24], who found for *C. miqueliana*, a species for tropical rain forest from southern Veracruz, that precipitation in the driest month was also a variable that explains the distribution of the species.

According to the Pleistocene refuge hypothesis, in places where climatic changes were not so extreme, many species would find refuge and be restricted to these areas [49]. This is consistent with those observed in this study since, using the distribution models; we observed an effect on the distribution of *Ceratozamia* species due to past climatic changes in the Quaternary (LGM). Specifically, we observed that montane species (*C. morettii* and *C. tenuis*) encountered mountain cloud forest conditions in highly restricted areas of the Chiconquiaco mountain range. According to the mountain humid forest model, they moved to lowlands in the LGM [50]. A population movement towards lowlands from highlands was expected since

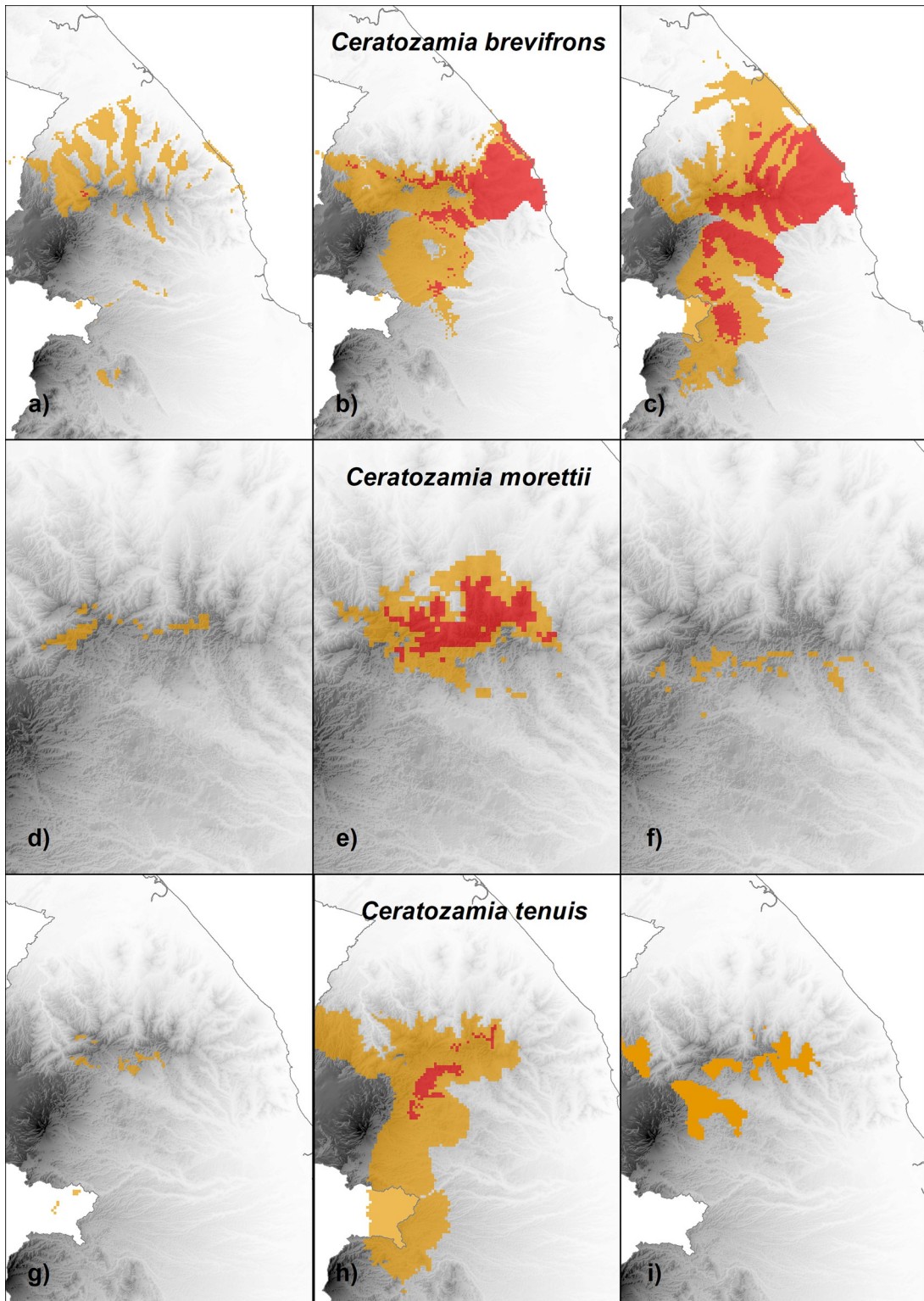

**Fig 4.** Potential distribution of *Ceratozamia brevifrons* a) in the past; b) present; c) future; *C. morettii* d) in the past; e) present; f) future; and *C. tenuis* g) in the past; h) present; i) future. Red: high suitability; Orange: medium.

**Table 5. Percentage of change between the different periods of the studied Tepezmaite cycads.**

| Change | C. brevifrons | C. morettii | C. tenuis |
|---|---|---|---|
| Past to present | 155% | 985% | 4005% |
| Present to future | 60% | -88% | -83% |

that, during the LGM, these lowlands would serve as refuges with limited habitat suitability, all this under the Pleistocene refuge hypothesis, which postulates climate-driven expansions and contractions of species ranges [50]. We found support for that hypothesis, observing that the populations of the three *Ceratozamia* species had limited habitat suitability during the LGM.

The above was more marked in the species *C. morettii* and *C. tenuis* under the "long-term *in situ* persistence" hypothesis because the mountain cloud forest currently has a fragmented distribution in our study region [48]. According to this hypothesis, *Ceratozamia* populations would be geographically structured and persistent (multiple glacial refuges), with no signs of demographic expansion before LGM and limited gene flow [48]. Comparable results of local persistence and numerous glacial refuges have been found for several species [48, 51–53]. In the case of the populations of *Ceratozamia* in the lowlands (*C. brevifrons*), we found that the species took refuge in the foothills of the Sierra de Chiconquiaco during glacial cycles.

Evidence suggests that *Ceratozamia* species found refuges in southern Mexico due to the climate changes of the Pleistocene, where the greatest number of basal clades and diversity of the genus are currently concentrated [49]. Though no Pleistocene refuges have been found north of the Trans-Mexican Volcanic Belt [49], showing that *Ceratozamia* expanded from southern Mexico to the north, so the species of the Trans-Mexican Volcanic Belt are of recent speciation [9, 49, 54]. Therefore, the diversification of *Ceratozamia* clades in the Pleistocene occurred in the mountainous region of the Sierra Madre Oriental, southern Mexico, and even in Central America [9]. The stage of climatic changes that influenced the distributions of the cycad species in this study, whose data come from the late Pleistocene, coincides with the origin of some species of the "Mexicana" clade, including *C. tenuis*, whose origin dates to the middle Pleistocene (1.4 Myr) [9]. Therefore, the reductions in the distribution of the three species that were recently separated at that time contributed to the isolation of their populations, avoiding subsequent interactions, which continue to the present since the three species are in different ecoregions.

Furthermore, with the information generated through the ecological niche models, we inferred a reduction in habitat suitability in the past of at least these three *Ceratozamia* species studied concerning the present. According to the Pleistocene refuge hypothesis, these species in the postglacial periods had an increase in their distribution up to the present time and are developing in secondary contact areas [55]. However, these cycad species did not undergo subsequent moments of encounter due to the presence of significant orographic barriers inherent to the Sierra de Chiconquiaco. These barriers are marked by rugged terrain, effectively isolating their current distributions. This scenario aptly illustrates allopatric speciation.

Currently, remnants of populations are concentrated in ravines with difficult access where agriculture and livestock activities are not profitable in Veracruz. Even under these conditions, the reduction of its population is continuous, although on a smaller scale due to livestock activities (e.g., goats and cows) and seasonal crops on a smaller scale. According to the evidence, we recognized that what remains today are relict populations of species whose lineage has developed through millions of years, enduring different climatic events of different intensities [26], placing them in various international and Mexican protection schemes [4, 56].

In addition to the above, we showed that future modeling indicates that their surface will be reduced to such a degree that they will practically disappear, especially for *C. morettii* and *C.*

*tenuis*. In the case of *C. miqueliana*, future modeling predicts no change in the suitability of its niche (Carvajal-Hernández, unpubl.). However, *C. miqueliana* is a lowland species subjected to summer temperatures exceeding 35˚C [24]. This contrasts with what is shown in the models of *C. morettii* and *C. tenuis*, which are found in mountainous regions with temperate temperatures. Still, it matches the future forecast of *C. brevifrons*, whose model indicates an increase in distribution.

Some models indicate that in Mexico, temperate ecosystems (coniferous forest, *Quercus* Forest, and mountain cloud forest) will suffer a reduction due to climate change [57, 58]. This coincides with our results since the species estimated to have a drastic reduction are precisely those that inhabit temperate forests (*C. morettii* and *C. tenuis*). This situation has been described with other temperate species, attributing changes in the distribution associated with the loss of humidity, assuming that according to their physiological tolerances, the species will be affected in the short term by a reduction in moisture rather than by a temperature increase [59]. In contrast, dry tropical ecosystems will have an increase in their distribution [59–61]. For this reason *C. brevifrons*, thriving in higher temperatures and elevated drought risks, is projected to expand its distribution. Other conifer species within Mexican montane forests along the Trans-Mexican Volcanic Belt reveal divergent responses to future climatic conditions. *Pinus hartwegii* and *Abies religiosa* are projected to suffer significant distribution losses due to elevational shifts by 2060, while *Pinus oocarpa* highlights potential niche space gains at lower elevations [60].

Climate change models indicate that tropical mountains will be the systems most affected due to the loss of species [62], and therefore contained biodiversity is higher than lowlands in biodiverse tropical zones [63]. It is estimated that the mountains will become warmer and drier, with a trend towards greater drought in the lower mountainous areas, which will cause changes in the distribution of plant species, some will benefit, but others will be harmed in various aspects [64, 65]. In some mountains of the world, the ranges of plant species distribution will be shortened due to climatic changes, droughts in low areas, and extreme snowfall in high regions [65]. These insights emphasize the need for human intervention to facilitate altitudinal migration, ensuring conifer populations align with changing climates [59]. More than traditional conservation measures may be needed in such dynamic circumstances to preserve current forest compositions.

In light of the impending challenges posed by the effects of climate change on *C. morettii* and *C. tenuis*, we emphasize the need to take action now to preserve these microendemic species. Building upon the insights gathered from our study, we recognize the need for more initiative-taking measures to ensure the survival of these species in their restricted habitats [66]. Recognizing the intricate interplay between climate change and habitat degradation, we emphasize the importance of habitat restoration initiatives [67]. By rehabilitating and expanding the suitable habitats for these species, we can create more resilient ecosystems capable of withstanding the challenges posed by changing climatic conditions.

In addition, successful *in-situ* and *ex-situ* breeding programs in Mexico have shown that it is possible to relocate multiple individuals of cycads [68]. Using species distribution models and *ex-situ* breeding, we can conduct relocation experiments and monitor individual progress in areas suitable for current and future habitats. We are also recommended to continue to protect these species and their habitats by creating protected natural areas, including local and international government protection schemes, and updating the risk statuses of species already under protection [67]. Furthermore, establishing protected areas and conservation corridors specific to these *Ceratozamia* species is paramount [69]. These designated zones could safeguard their natural habitats and enable controlled management and monitoring, ensuring long-term survival [66].

## Conclusion

We consider that the three species of cycads are highly threatened due to their limited distribution area, making them microendemic. Past climate models indicate isolation of their populations, the same that continues to the present. Through future climate models, a drastic reduction in the suitability of the habitat of *C. morettii* and *C. tenuis* that currently inhabit temperate mountainous regions are expected to increase their current vulnerability. On the contrary, an increase in distribution is estimated for *C. brevifrons* that now live in conditions of higher temperature and drought. These models show the trajectory of the species through time, observing that they have endured different climatic changes that affect their distribution. However, future models indicate that two species will be susceptible to disappearing quickly due to climatic trends affecting their habitat expected to continue in their distribution area. This raises different scenarios for biodiversity in general.

Cycads represent interesting species from a biogeographic perspective due to their restricted distribution, but for that reason, they also represent a challenge for their conservation. Understanding and recognizing the factors and patterns that originate biodiversity on temporal and spatial scales will help the conservation strategies of ecosystems and species succeed. Unfortunately, how habitats and species can respond to climate change has yet to be fully discovered, so there is still an urgent and challenging task to fill these information gaps. Therefore, to guide conservation efforts efficiently and in future ecological studies, multiple perspectives and approaches are necessary to be integrated (for example, propagation and cultivation of these species, the establishment of protected natural areas, and relocation experiments in suitable zones), as well as to know the complete history of the groups to be studied as in this study.

## Acknowledgments

We thank Biologist Hector Hernández Andrade for his permission to conduct fieldwork on his property and Merbin Tornero Conde for his support in the field.

## Author Contributions

**Conceptualization:** Jorge Antonio Gómez-Díaz, César Isidro Carvajal-Hernández, Wesley Dáttilo.

**Data curation:** Jorge Antonio Gómez-Díaz, César Isidro Carvajal-Hernández.

**Formal analysis:** Jorge Antonio Gómez-Díaz.

**Funding acquisition:** Jorge Antonio Gómez-Díaz.

**Investigation:** César Isidro Carvajal-Hernández, Wesley Dáttilo.

**Writing – original draft:** Jorge Antonio Gómez-Díaz.

**Writing – review & editing:** Jorge Antonio Gómez-Díaz, César Isidro Carvajal-Hernández, Wesley Dáttilo.

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
