## [Decision Letter · Decision Letter 0]

14 Jul 2023

PONE-D-23-08499Past, present and future in the geographical distribution of Mexican tepezmaite cycads: genus CeratozamiaPLOS ONE

Dear Dr. Carvajal-Hernández,

Thank you for submitting your manuscript to PLOS ONE. After careful consideration, we feel that it has merit but does not fully meet PLOS ONE’s publication criteria as it currently stands. Therefore, we invite you to submit a revised version of the manuscript that addresses the points raised during the review process.

We look forward to receiving your revised manuscript.

Kind regards,

Marcela Pagano, Ph.D, M.D.

Academic Editor

PLOS ONE

“The research was supported by Mohamed bin Zayed Species Conservation Fund (project number 192521089) and also the Centro de Investigaciones Tropicales, Universidad Veracruzana for their support for conducting fieldwork, awarded to JAGD.”

“We thank Mohamed bin Zayed Species Conservation Fund and the Centro de Investigaciones Tropicales (CITRO), Universidad Veracruzana for their support for conducting fieldwork. We also thank Biologist Hector Hernández Andrade for his permission to carry out fieldwork on his property and to Merbin Tornero Conde for his support in the field.”

“The research was supported by Mohamed bin Zayed Species Conservation Fund (project number 192521089) and also the Centro de Investigaciones Tropicales, Universidad Veracruzana for their support for conducting fieldwork, awarded to JAGD.”

5. We note that Figure 1 in your submission contain copyrighted images. All PLOS content is published under the Creative Commons Attribution License (CC BY 4.0), which means that the manuscript, images, and Supporting Information files will be freely available online, and any third party is permitted to access, download, copy, distribute, and use these materials in any way, even commercially, with proper attribution. For more information, see our copyright guidelines: http://journals.plos.org/plosone/s/licenses-and-copyright.

6. We note that Figures 2, 3 and 4 in your submission contain [map/satellite] images which may be copyrighted. All PLOS content is published under the Creative Commons Attribution License (CC BY 4.0), which means that the manuscript, images, and Supporting Information files will be freely available online, and any third party is permitted to access, download, copy, distribute, and use these materials in any way, even commercially, with proper attribution. For these reasons, we cannot publish previously copyrighted maps or satellite images created using proprietary data, such as Google software (Google Maps, Street View, and Earth). For more information, see our copyright guidelines: http://journals.plos.org/plosone/s/licenses-and-copyright.

1. You may seek permission from the original copyright holder of Figures 2, 3 and 4 to publish the content specifically under the CC BY 4.0 license. 

Reviewers' comments:

Reviewer's Responses to Questions

**Comments to the Author**

1. Is the manuscript technically sound, and do the data support the conclusions?

Reviewer #1: Yes

2. Has the statistical analysis been performed appropriately and rigorously? 

Reviewer #1: Yes

3. Have the authors made all data underlying the findings in their manuscript fully available?

Reviewer #1: Yes

4. Is the manuscript presented in an intelligible fashion and written in standard English?

Reviewer #1: Yes

5. Review Comments to the Author

Reviewer #1: General comments.

Interesting paper about three rare, endemic, of restricted distribution, (and beautiful ! ) Ceratozamia species.

Some clarifications and improvements are needed. See my specific comments.

Having all this knowledge about these three species, and making a good point about the expected impacts of climatic change, I believe that the authors are in the position to suggest some specific actions for improving its conservation in the Era of climatic change. They mention that superficially on Conclusion, but I believe it would be worth to develop that on Discussion Section

Specific comments

Abstract.

Line 39. “the tropical climate species will apparently be favored”. Say which specie: “the tropical climate species (C. ???) will apparently be favored.

Introduction

L. 52. “after Australia with 80% endemic species of the country”. It is not clear. Is it: “after Australia with 80% endemic species of the world? Or do you mean Mexico has as endemic 80 % of the Cycads that distribute in Mexico? Please clarify.

68-69. The three species receive the same common/local name of ““Tepezmaite”? Reword to clarify that.

73. “C. tenuis has not been evaluated by Mexican laws”. I do not believe that “laws” are able to evaluate. The species could be or not protected by the Mexican laws. Perhaps to say “C. tenuis has not been evaluated by Mexican federal agency for the environment protection” or so.

79-80: Better (I guess what you mean): “The three species grow on humid mountain pine-oak forest”. Or do you mean mountain cloud forest? Because the cloud forest is among the most endangered biome in Mexico by climatic change, combined by its reduced distribution. See Table 4 on Rehfeldt et al 2012. Ecological Applications, 22(1), pp. 119–141.

103. “200 ma” ¿ what do you mean with “ma” ? millions of years? Would it be “Myr”? Perhaps better do not abbreviate to prevent confusion.

108. (Same correction for line 121 and so for) In English “nombres propios” are capitalized. It should be “Tepezmaite’s”, like we use “Douglas-fir” and not “douglas-fir”.

Methods

158. Comma needed on: “C. brevifrons remained, 16 of C. morettii and..”

173-176. And perhaps because so far RCP 8.5 appears to be the most likely scenario, given the steady trend so far of green-house effect gas emissions? See for example: http://www.columbia.edu/~jeh1/Documents/PipelinePaper.2023.05.19.pdf

Results

212. Better: “exceptionally small distribution área”

215-218. I believe it is better if you say what are the relevant variables, by its name, and not by its code name. As you did on 290-297.

232. A “a slight decrease” … regarding present distribution?

Figure 4,a lthoug key for the argument of the manuscript, it is of poor resolution and difficult to appreciate the details. Each panel is very small. Inbelieve you can zoom a bit, eliminating not relevant areas (with no specie distribution) both at north and at south, to enhance the species distribution area on each panel.

Table 5 and references to that on the main text. I do not see the point to estimate “past to future”. What is the benefit of that? Is meaningless. The relevant is present to future, and it is interesting past to present. That is it.

Discussion

345-346. say some examples of what you rfer to as “notable orographic

barriers” A specific river or mountain or volcano?

366-369. In addition to your reference [55], see Rehfeldt et al 2012 Ecol. Appl.

372-374. Perhaps say that such a catastrophic predicted reduction of P. oocarpa is specifically for Veracruz, because P. oocarpa, exactly because it is on of the species that grows in the low altitudinal limit of the pine-oak forest, is apt to site dryer than for most of the pines. That make that it is a pine that loss less area, compared with other Mexican pines. See for example Gómez-Pineda et al 2020, Ecological Applications, 30(2), e02041. https://esajournals.onlinelibrary.wiley.com/doi/10.1002/eap.2041

390. what do you mean with “apparent” droughts? Or do you mean sporadic droughts?

Conclusions

408. I do not believe that you focused on your research on “current anthropogenic pressures” enough to put that on Conclusions. To mention that on Introduction or so it is OK, but your modeling was done solely with climatic variables, was it not?

424. “.. multiple perspectives and approaches are necessary to be

integrated” OK, like what approaches are needed? Examples? Suggestions? Do that on Discussion.

6. PLOS authors have the option to publish the peer review history of their article (what does this mean?). If published, this will include your full peer review and any attached files.

Reviewer #1: No

---

## [Author Response · Author response to Decision Letter 0]

29 Aug 2023

Dear Editor,

With this letter, we respond to the comments provided by the reviewers regarding our manuscript titled "Past, present and Future in the geographical distribution of Mexican tepezmaite cycads: genus Ceratozamia." It is important to note that all the changes suggested by the reviewer have been fully addressed.

1.We have addressed the formatting requirements established by PLOS ONE by the style templates available in: (https://journals.plos.org/plosone/s/file?id=wjVg/PLOSOne_formatting_sample_main_body.pdf and https://journals.plos.org/plosone/s/file?id=ba62/PLOSOne_formatting_sample_title_authors_affiliations.pdf

2. We addressed the financial disclosure comment and indicated, "The funders had no role in study design, data collection and analysis, decision to publish, or preparation of the manuscript." Thus, the financial statement is now as follows:

Funding information

The research was supported by Mohamed bin Zayed Species Conservation Fund (project number 192521089) and also the Centro de Investigaciones Tropicales, Universidad Veracruzana for their support for conducting fieldwork. The funders had no role in study design, data collection and analysis, decision to publish, or preparation of the manuscript.

3. We removed everything related to the funding information from the Acknowledgments section. It now reads as follows:

Acknowledgments

We thank Biologist Hector Hernández Andrade for his permission to conduct fieldwork on his property and Merbin Tornero Conde for his support in the field.

4. The information obtained for the geographic distribution analyses reported in this study was based on publicly accessible records obtained from GBIF. This is mentioned in the methodology, and the references include links to these records and the dates of consultation. It now reads as follows:

To map the spatial distribution of species, we obtained records of the studied Ceratozamia species from our data in the field, data from the XAL, XALU, and MEXU herbaria, data from the literature and the Global Biodiversity Information Facility (GBIF), as well as corroboration of localities in the field. Only georeferenced records were chosen. We found 14 occurrences for C. brevifrons [35], 16 for C. morettii [36] and 26 for C. tenuis [37].

[35] GBIF.org. The Global Biodiversity Information Facility; 2023 [cited 21 March 2023). Database: figshare [Internet]. Available from: GBIF Occurrence Download https://doi.org/10.15468/dl.etfbpd

[36] GBIF.org. The Global Biodiversity Information Facility; 2023 [cited 21 March 2023). Database: figshare [Internet]. Available from: GBIF Occurrence Download https://doi.org/10.15468/dl.2smk9x

[37] GBIF.org. The Global Biodiversity Information Facility; 2023 [cited 21 March 2023). Database: figshare [Internet]. Available from: GBIF Occurrence Download https://doi.org/10.15468/dl.pp2gf2

5. The six photographs appearing in Figure 1 were taken directly by the authors and depict individuals of the studied species found in their natural habitat. Therefore, the authors are in agreement that the panel of Figure 1 should be made available freely online, and we have no objection to their use by a third party.

6. The three maps have been modified, and the versions presented in this new submission were generated in ArcGIS Pro based on freely accessible maps available on the website of the National Institute of Geography and Statistics (Instituto Nacional de Geografía y Estadística, INEGI), which is a government agency of Mexico. The maps are freely available at the following link: https://www.inegi.org.mx/app/geo2/elevacionesmex/

7. We reviewed the references and they were arranged according to the guidelines for PLOS ONE authors.

We provide our responses to each of the suggestions made by the reviewer.

Best regards,

The Authors

Xalapa, Veracruz, 16/08/2023

Reviewer #1: General comments

Rev. Interesting paper about three rare, endemic, of restricted distribution (and beautiful!) Ceratozamia species.

Some clarifications and improvements are needed. See my specific comments.

Having all this knowledge about these three species, and making a good point about the expected impacts of climatic change, I believe that the authors are in the position to suggest some specific actions for improving its conservation in the Era of climatic change. They mention that superficially on Conclusion, but I believe it would be worth to develop that on Discussion Section

Answer: We wholeheartedly agree with your suggestion about proposing specific actions to conserve these species in the face of climatic change. As you rightly pointed out, our knowledge of these three species and their potential vulnerability to climate change positions us to contribute meaningfully to their conservation. While we briefly touch upon this aspect in our Conclusion, we acknowledge the merit of further elaborating upon it in the Discussion Section. In response to your insightful recommendation, we will expand upon the potential conservation strategies in the Discussion Section. By providing a dedicated space for this discussion, we aim to explore the practical and actionable steps that can be taken to safeguard these Ceratozamia species in the era of climatic change. This enhancement will bolster the significance of our research and offer valuable insights to conservation practitioners and policymakers. The paragraph that was added to the discussion section regarding potential actions for the conservation of species that will be most impacted by climate change is (L 411-430):

“In light of the impending challenges posed by the effects of climate change on C. morettii and C. tenuis, we emphasize the need to take action now to preserve these microendemic species. Building upon the insights gathered from our study, we recognize the need for more initiative-taking measures to ensure the survival of these species in their restricted habitats [66]. Recognizing the intricate interplay between climate change and habitat degradation, we emphasize the importance of habitat restoration initiatives [67]. By rehabilitating and expanding the suitable habitats for these species, we can create more resilient ecosystems capable of withstanding the challenges posed by changing climatic conditions. 

In addition, successful in-situ and ex-situ breeding programs in Mexico have shown that it is possible to relocate multiple individuals of cycads [68]. Using species distribution models and ex-situ breeding, we can conduct relocation experiments and monitor individual progress in areas suitable for current and future habitats. We are also recommended to continue to protect these species and their habitats by creating protected natural areas, including local and international government protection schemes, and updating the risk statuses of species already under protection [67]. Furthermore, establishing protected areas and conservation corridors specific to these Ceratozamia species is paramount [69]. These designated zones could safeguard their natural habitats and enable controlled management and monitoring, ensuring long-term survival [66].” 

Rev. Line 39. “the tropical climate species will be favored”. Say which specie: “the tropical climate species (C. ???) will apparently be favored.

Answer: Done. (L 46-47)

On the other hand, the tropical climate species (C. brevifrons) will be favored.

Rev. L. 52. “after Australia with 80% endemic species of the country”. It is not clear. Is it: “after Australia with 80% endemic species of the world? Or do you mean Mexico has as endemic 80 % of the Cycads that distribute in Mexico? Please clarify.

Answer: Thank you for your query regarding the statement in line 52 of our paper. We apologize for any confusion. The intended clarification is that from the total of species from Mexico approximately 80% are endemic to Mexico. The sentence was modified as follows (L 57-60):

In Mexico, 74 species of cycads are recognized in three genera: Ceratozamia, Dioon, and Zamia [2]. Of these species, 80% are endemic to Mexico, making it the second most diverse country in the world regarding cycad diversity, after only Australia [3].

Rev. 68-69. The three species receive the same common/local name of ““Tepezmaite”? Reword to clarify that.

Answer: This clarification was made in the text (L74-77):

Within the genus, Ceratozamia are C. brevifrons (Fig 1a), C. morettii (Fig 1b), and C. tenuis (Fig 1c), three micro endemic species from central Veracruz, known collectively by the local name “Tepezmaite” or “Palmita.”

Rev. 73. “C. tenuis has not been evaluated by Mexican laws”. I do not believe that “laws” are able to evaluate. The species could be or not protected by the Mexican laws. Perhaps to say “C. tenuis has not been evaluated by Mexican federal agency for the environment protection” or so.

Answer: We agree with the reviewer's comment, and the wording has been revised as follows (L 81-83):

Mexican laws do not protect C. tenuis due to its recent lectotype assignment in 2016 [14], previously considered a synonym of C. mexicana.

Rev. 79-80: Better (I guess what you mean): “The three species grow on humid mountain pine-oak forest”. Or do you mean mountain cloud forest? Because the cloud forest is among the most endangered biome in Mexico by climatic change, combined by its reduced distribution. See Table 4 on Rehfeldt et al 2012. Ecological Applications, 22(1), pp. 119–141.

Answer: The wording was modified to clarify the type of vegetation each species inhabits (L 83-85).

Ceratozamia morettii and C. tenuis grow in mountain cloud forests, while C. brevifrons grow in tropical oak forests; both environments are among Mexico's most threatened types of vegetation [15].

Rev. 103. “200 ma” ¿ what do you mean with “ma” ? millions of years? Would it be “Myr”? Perhaps better do not abbreviate to prevent confusion.

Answer: The change was made (L 111)

Rev. 108. (Same correction for line 121 and so for) In English “nombres propios” are capitalized. It should be “Tepezmaite’s”, like we use “Douglas-fir” and not “douglas-fir”.

Answer: Done. The change was made throughout the entire document.

Rev. 158. Comma needed on: “C. brevifrons remained, 16 of C. morettii and..”

Answer: Done.

Rev. 173-176. And perhaps because so far RCP 8.5 appears to be the most likely scenario, given the steady trend so far of green-house effect gas emissions? See for example: http://www.columbia.edu/~jeh1/Documents/PipelinePaper.2023.05.19.pdf

Answer: We agree; we have added the reviewer's observation (L 183-186):

We only considered the RCP 8.5 (936 ppm of CO2) scenario because it represents a continuous increase in global emissions throughout the 21st century and appears to be the most likely scenario, given the steady trend so far of greenhouse effect gas emissions [42].

Rev. 212. Better: “exceptionally small distribution área”

Answer: Change made (L 222-223):

…and having an exceptionally small distribution area in central Veracruz (Fig 3),

Rev. 215-218. I believe it is better if you say what are the relevant variables, by its name, and not by its code name. As you did on 290-297.

Answer: Done. The full names of the variables have been incorporated (L 226-233):

In the case of C. brevifrons, the most important variables were temperature seasonality (Bio 4,41%), mean diurnal range (Bio2,17%), and precipitation of wettest month (Bio 13,16%); for C. morettii, the most important variables were mean temperature of wettest quarter (Bio 8, 29%), Bio 13 (24%) and Bio 4 (23%), and for C. tenuis were precipitation of driest month (Bio 14,29%), mean temperature of coldest quarter (Bio 11,24%) and precipitation of warmest quarter (Bio 18,20%; Table 3). The only variables shared for the three species are Bio 13 and Bio 18.

Rev. 232. A “a slight decrease” … regarding present distribution?

Answer: The clarification has been made in the text (L 246-247):

The past model suggested a slight decrease regarding the present distribution (Table 4)

Rev. Figure 4, although key for the argument of the manuscript, it is of poor resolution and difficult to appreciate the details. Each panel is very small. I believe you can zoom a bit, eliminating not relevant areas (with no specie distribution) both at north and at south, to enhance the species distribution area on each panel.

Answer: We appreciate your feedback regarding Figure 4. Your observation about each panel's resolution and small size is duly noted. We completely agree that enhancing the figure's clarity and details is essential for effectively supporting the manuscript's argument. Your suggestion to zoom in on the relevant distribution areas and eliminate non-relevant sections is well-founded. We optimized the presentation of each panel in Figure 4. This enhancement will undoubtedly contribute to a better understanding and appreciation of species distribution. 

Rev. Table 5 and references to that on the main text. I do not see the point to estimate “past to future”. What is the benefit of that? Is meaningless. The relevant is present to future, and it is interesting past to present. That is it.

Answer: We agree with the reviewer and have removed that information.

Rev. 345-346. say some examples of what you rfer to as “notable orographic

Barriers” A specific river or mountain or volcano?

Answer: The wording was altered as follows (L 362-366):

However, these cycad species did not undergo subsequent moments of encounter due to the presence of significant orographic barriers inherent to the Sierra de Chiconquiaco. These barriers are marked by rugged terrain, effectively isolating their current distributions. This scenario aptly illustrates allopatric speciation.

Rev. 366-369. In addition to your reference [55], see Rehfeldt et al 2012 Ecol. Appl.

Answer: We include the citation proposed by the author as we deem it relevant to the presented idea. (L 384-386):

Some models indicate that in Mexico, temperate ecosystems (coniferous forest, Quercus Forest, and mountain cloud forest) will suffer a reduction due to climate change [57,58].

Rev. 372-374. Perhaps say that such a catastrophic predicted reduction of P. oocarpa is specifically for Veracruz, because P. oocarpa, exactly because it is on of the species that grows in the low altitudinal limit of the pine-oak forest, is apt to site dryer than for most of the pines. That make that it is a pine that loss less area, compared with other Mexican pines. See for example Gómez-Pineda et al 2020, Ecological Applications, 30(2), e02041. https://esajournals.onlinelibrary.wiley.com/doi/10.1002/eap.2041

Answer: We have replaced the originally provided citation with the one recommended by the reviewer, considering it more appropriate. Additionally, the wording has been adjusted to ensure the presented idea is clear. The revised wording is as follows (L 394-398):

Other conifer species within Mexican montane forests along the Trans-Mexican Volcanic Belt reveal divergent responses to future climatic conditions. Pinus hartwegii and Abies religiosa are projected to suffer significant distribution losses due to elevational shifts by 2060, while Pinus oocarpa highlights potential niche space gains at lower elevations [60].

Rev. 390. what do you mean with “apparent” droughts? Or do you mean sporadic droughts?

Answer: We adjusted the wording to avoid confusion (L 384-388).

Some models indicate that in Mexico, temperate ecosystems (coniferous forest, Quercus Forest, and mountain cloud forest) will suffer a reduction due to climate change [57,58]. This coincides with our results since the species estimated to have a drastic reduction are precisely those that inhabit temperate forests (C. morettii and C. tenuis).

Rev. 408. I do not believe you focused on your research on “current anthropogenic pressures” enough to put that on Conclusions. To mention that on Introduction or so it is OK, but your modeling was done solely with climatic variables, was it not?

Answer: We agree with the reviewer and have removed the mention of anthropogenic pressures, as our study is focused on climatic factors rather than anthropogenic pressures (L433-445):

We consider that the three species of cycads are highly threatened due to their limited distribution area, making them microendemic. Past climate models indicate isolation of their populations, the same that continues to the present. Through future climate models, a drastic reduction in the suitability of the habitat of C. morettii and C. tenuis that currently inhabit temperate mountainous regions are expected to increase their current vulnerability. On the contrary, an increase in distribution is estimated for C. brevifrons that now live in conditions of higher temperature and drought. These models show the trajectory of the species through time, observing that they have endured different climatic changes that affect their distribution. However, future models indicate that two species will be susceptible to disappearing quickly due to climatic trends affecting their habitat expected to continue in their distribution area. This raises different scenarios for biodiversity in general.

Rev. 424. “.. multiple perspectives and approaches are necessary to be

integrated” OK, like what approaches are needed? Examples? Suggestions? Do that on Discussion.

Answer: Two paragraphs were added to the discussion on this topic. Furthermore, the wording of this sentence in the conclusion was modified (L 411-430).

Therefore, to guide conservation efforts efficiently and in future ecological studies, multiple perspectives and approaches are necessary (For example, propagation and cultivation of these species, the establishment of protected natural areas, and relocation experiments in suitable zones) to be integrated, as well as to know the complete history of the groups to be studied as in this study.

In light of the impending challenges posed by the effects of climate change on C. morettii and C. tenuis, we emphasize the need to take action now to preserve these microendemic species. Building upon the insights gathered from our study, we recognize the need for more initiative-taking measures to ensure the survival of these species in their restricted habitats [66]. Recognizing the intricate interplay between climate change and habitat degradation, we emphasize the importance of habitat restoration initiatives [67]. By rehabilitating and expanding the suitable habitats for these species, we can create more resilient ecosystems capable of withstanding the challenges posed by changing climatic conditions. 

In addition, successful in-situ and ex-situ breeding programs in Mexico have shown that it is possible to relocate multiple individuals of cycads [68]. Using species distribution models and ex-situ breeding, we can conduct relocation experiments and monitor individual progress in areas suitable for current and future habitats. We are also recommended to continue to protect these species and their habitats by creating protected natural areas, including local and international government protection schemes, and updating the risk statuses of species already under protection [67]. Furthermore, establishing protected areas and conservation corridors specific to these Ceratozamia species is paramount [69]. These designated zones could safeguard their natural habitats and enable controlled management and monitoring, ensuring long-term survival [66].

---

## [Editor Report · Decision Letter 1]

11 Jan 2024

Past, present and future in the geographical distribution of Mexican Tepezmaite cycads: genus Ceratozamia

PONE-D-23-08499R1

Dear Dr. César I. Carvajal-Hernández,

We’re pleased to inform you that your manuscript has been judged scientifically suitable for publication and will be formally accepted for publication once it meets all outstanding technical requirements.

Kind regards,

Marcela Pagano, Ph.D, M.D.

Academic Editor

PLOS ONE
---

## [Editor Report · Acceptance letter]

31 Jan 2024

PONE-D-23-08499R1 

PLOS ONE

Dear Dr. Carvajal-Hernández, 

I'm pleased to inform you that your manuscript has been deemed suitable for publication in PLOS ONE. Congratulations! Your manuscript is now being handed over to our production team.

Kind regards, 

on behalf of

Dr. Marcela Pagano 

Academic Editor

PLOS ONE